# There's a Basilisk in the Bathwater: AI and the Apocalyptic Imagination

## Avery Isbrücker [1,2]

1  Queen's School of Religions, Queen's University, Kingston, ON K7L 3N6, Canada; avery.isb@gmail.com
2  Department of English, Concordia University, Montréal, QC H3G IMB, Canada

**Abstract:** Deciding what to make of secular, religious, and spiritual speculations about AI and digital technologies can be overwhelming, and focusing on the extreme utopic or dystopic outcomes may be obscuring the larger facts. Is this technology a beautiful blessing or a damning curse? What can paying close attention to these technologies and the discourse surrounding them show? How founded are our anxieties? By following the apocalyptic throughline in this rhetoric across fields in recent years, this essay seeks to consider the effect of apocalyptic thought on recent developments in tech, and consider how this worldview orients our future. The deterministically utopic, dystopic, and apocalyptic rhetoric surrounding these technologies obscures their function and efficacy, giving agency to what is functionally still just a tool, the use for which depends on its designers and users.

**Keywords:** apocalypticism; AI; transhumanism; Superintelligence; Singularity; enchanted determinism; Roko's Basilisk; speculation; eschatology; algorithmic spirituality

## 1. Introduction

Developments in the field of AI and machine learning systems are often discussed as developments towards a powerful, prescient being with the potential to reshape the world (Anderson and Rainie 2023; Singler 2019; Crawford 2021a; Mercer and Trothen 2021a). Current AI (narrow AI, machine learning and deep learning systems) have yet to reach a point we would consider to be AGI (Artificial General Intelligence, or Strong AI). Still, our speculation about—and immersion into—these new technologies continues.

AGI would match human performance across fields, responsibilities, and measures of intelligence (Mercer and Trothen 2021a). After AGI, we may develop Superintelligence. This machine would surpass all of humanity in regard to its understanding of human history, morals, emotions, intelligence, as well as a superhuman capacity for pattern recognition and problem-solving (Kurzweil 2022). Superintelligence might be omnipresent in a way analogous to the Cloud (Mercer and Trothen 2021c, p. 187). With a wide reach over technological networks, these machines have the potential for omniscience and omnipotence as well. With these godlike abilities, the development of Superintelligence may be seen as the next step in a spiritual and scientific evolution for our species (Mercer and Trothen 2021a; Calheiros 2014; Kurzweil 2022).

Designing such a system might be seen as humanity "reaching for something transcendent beyond our finite and fallible selves", towards a kind of "technological divinity" (Mercer and Trothen 2021a, p. 188). In the best of cases, this superintelligent machine would have humanity's best interests at heart. In the worst, future AI may spell ruin for the world.

This essay considers the reification of AI as an autonomous, spectral figure, whose eventual development into a form of Superintelligence seems inevitable. These efforts are often based in eschatological visions of digital technology, which spiritualizes the seemingly immaterial digital plane (Calheiros 2014; Naudé 2021; Singler 2020; Cotter et al. 2022). Biblically, eschatology is connected to the Apocalypse, referring to the (often theological)

study of the final destiny of humankind, in body, mind, and soul. It is connected to death, judgement, the second coming, and resurrection. Most work on the subject in religious studies has been focused on Christianity and Christian responses. This essay follows this tradition, and pays particular attention to the secular transhuman movement, which has been previously discussed as a an inheritor of Christian thought (Cole-Turner 2015). If we are to continue down the path of designing digital technologies based on ourselves, we should be very aware of the intelligent design behind the veil of these systems—lest we fool ourselves into believing a clever charade.

## 2. The All-Seeing Algorithm and the Eschatological Trend in Transhuman Thought

Far from being what we would consider AGI, current AI focus largely on algorithms and data rather than the historic attempts of symbolic logic (Naudé 2021). The personal experience of algorithmically powered media gives the impression that these systems "know us deeply", giving them a "sublime quality in the public imagination" (Cotter et al. 2022, p. 2915). Users of popular platforms like TikTok, YouTube and Instagram interact with algorithms in ways which seem conducive to religious uses and ecstatic visions (Somer et al. 2021). This has led to algorithms being seen through spiritual and theistic lenses (Cohen 2023; Singler 2020; Cotter et al. 2022)[1].

Fast-paced "algorithmically powered media" like TikTok, Tinder, and Instagram are what Dr. Aris Komporozos-Athanasiou calls speculative technologies: "commodified digital infrastructures [which] enabl[e] the circulation of speculative imaginations... key nodes for the generation of data and images that both represent and occlude the uncertain conditions of everyday social life" (Komporozos-Athanasiou 2022, p. x). Komporozos-Athanasiou's 'speculation' is a kind of 'connection', a preemptive (and often tactical) endorsement of uncertainty with the goal of "social survival". Komporozos-Athansiou's speculation further encompasses how we make our speculations "actionable in the present", as they operate in a state of simultaneous imminence and incertitude (Komporozos-Athanasiou 2022, p. 3).

These platforms have been studied in connection to a rise in conspiratorial and con-spiritual thought (Cotter et al. 2022; Cohen 2023). In their 2022 article In FYP We Trust, Cotter et al. refer to this phenomenon as 'algorithmic conspirituality' which relies on an algorithm's

> "... capacity to find unlikely connections across massive data sets about individuals. This capacity often results in eerily relevant content recommendations that seem to be designed specifically for individuals seeing them. [...] When algorithmically curated content resonates deeply with viewers and reaches them at the right moment, the experience may be read as divine, giving the impression of a powerfully all-seeing algorithm" (Cotter et al. 2022, p. 2924).

The desire to see machine learning systems like an oracle or a prescient entity demonstrates the belief that algorithms can unveil to users "things about themselves that they cannot see" (Cotter et al. 2022, p. 2913). This illusion is caused by the program's construction of what John Cheney Lippold calls "an algorithmic identity"; a facsimile of the user, based on quantifiable metrics like "their race, age, gender, or socioeconomic class" (Simpson et al. 2022, p. 2). This information is then used to curate users' feeds, presenting them with material which the algorithm determines their demographic would like. Deep learning AI systems rely on the assumption "that accurate prediction is fundamentally about reducing the complexity of the world", as they make predictions by flattening complex data fields into simplified categories (Crawford 2021b, p. 213). This can adversely affect users' routine experiences "by biasing personalization, reinforcing racism, or through their ability to determine relevance" (Simpson et al. 2022, p. 5).

The basis for algorithmic ways of knowing can be traced back to the 2013 Google Brain project which was headed by techno-utopian pioneer Ray Kurzweil (Calheiros 2014, p. 6). The purpose of the Google Brain was to anticipate a user's search before they did, by enacting a reflection of its assumption of who we are. Google Brain explores the

possibilities of technical applications for large-scale 'neural networks', and has contributed technological advancements to open-source and deep-learning software, natural language recognition, and computer-vision-assisted diagnosis in healthcare (Helms et al. 2018). The underlying ethos of the project falls in line with Kurzweil's Singularity, and the theory of global consciousness; seeing digital technologies as enabling a fusion of social and technical connections in a way which may ultimately generate "a spiritual connectedness then able to transcend humanity" (Calheiros 2014, p. 3).

Sociology researcher and specialist in transhumanism, Cecilia Calheiros studied the eschatological expectations within the discourse of cybernetics, which focuses mainly on the parallels to be found between technological and biological feedback processes (Calheiros 2014). The early development and ethos of cybernetics, combined with the Californian counter-culture movement of the 1960s and 1970s, gave rise to a kind of "techno-paganism" which remains "ever hungry for esotericism and technology" (Calheiros 2014, p. 2). The spread of the Internet in the 1980s and 1900s carried with it the belief that this technology would "transcend individual thoughts by creating a world community", in a step that is natural to our evolution (Calheiros 2014, p. 3).

This worldview is connected to the generation of technological innovations, which are "influenced by eschatological concerns, linked to the working of the spirit and to the ways to augment its power via techno-sciences" (Calheiros 2014, p. 2).

> "Called 'global consciousness', this belief rests upon in the notion that this medium would link individuals technically, but also spiritually. Such interconnection would spawn a collective spirit, some superior entity which would signal the advent of the "Spirit Realm"" (Calheiros 2014, p. 3).

This kind of collective spirit, or superior entity, may be seen as a form of Superintelligence, or Singularity.

The belief that technology can overcome the limits of the natural world, and take an active role in human evolution, is a transhuman one (Cole-Turner 2015; Calheiros 2014). Taking for granted our lives as fleshy, embodied beings, some believe that, by one day succeeding in uploading our minds, we might achieve immortality (Cole-Turner 2015; Singler 2019; Calheiros 2014). Christianity (an apocalyptic faith) has been compared to transhumanism in the shared ideal of seeking salvation beyond mortal flesh, which transhumanists may seek to do by "transferring the human spirit onto a machine" (Calheiros 2014, p. 6).

The achievement of these goals seems increasingly tenable, as progress in neurotechnology is making waves with companies like Neuralink Corps. Founded by Elon Musk and a small team of scientists in 2016, Neuralink Corps is in the process of developing implantable, brain–computer interfaces. As of 20 March 2024, Neuralink has reported one successful human patient, 29-year-old Noland Arbaugh (Adarlo 2024). Arbaugh is quadriplegic, and has stated that the implant has improved his quality of life significantly. Arbaugh is a believer in the future possibilities of this technology, saying it will change the world (CBC News 2024). Arbaugh's case seems to demonstrate a successful step toward the wider implementation of Neuralink's first goal to "[c]reate a generalized brain interface to restore autonomy to those with unmet medical needs"; however, it begs for further discussion of its second goal, to "unlock human potential" ("Neuralink" 2024). What exactly this potential is remains unknown, but its design is influenced by a line of thinking that has eschatological and esoteric roots.

## 3. Enchanted Determinism and Transhuman Apocalypticism

When faith is placed in deep-learning systems to make accurate predictions about the world, they become seen as too complex or powerful to control, beyond regulation or refusal. This is what AI scholar Kate Crawford and tech historian Alex Campolo refer to as enchanted determinism (Crawford 2021b, p. 213). Enchanted determinism imbues "an almost theological quality" to deep learning systems as their approaches are "often uninterpretable, even to the engineers who created them", and are, therefore, mired in a

kind of mystified misunderstanding of their function (Crawford 2021b, p. 214). It envisions AI as being "enchanted, beyond the known world, [and] deterministic in that they discover patterns that can be applied with predictive certainty to everyday life" (Crawford 2021b, p. 213).

Enchanted determinism has two branches: tech utopianism (AI is seen as a universal solution) and tech dystopianism, which views AI and algorithms as "negative" and "as though they are independent agents" (Crawford 2021b, p. 214). Elon Musk expresses both utopic and dystopic forms, stating that one of the reasons he wants to colonize Mars is his fear of a terminator-style AI apocalypse (Isaacson 2023; Milmo 2023). Both the dystopian and utopian narratives within enchanted determinism imagine a line of clear progress, a deterministic sense of time, and create a clear Us/Them binary vision of reality, imbuing AI with an aura of otherworldly power. In this way, they are fundamentally apocalyptic, as well as connected to a history of cybernetic philosophy that is inherently esoteric and eschatological (Calheiros 2014, p. 2).

It is important to note that the Apocalypse is not inherently negative, as it is often portrayed in a contemporary pop-cultural, or post-apocalyptic context. If on the side of the Elect, an Apocalypse is the ultimate rapture, in a biblical sense this can be seen as the ultimate entrance of all worthy faithful (living and dead) to Paradise (DiTommaso 2011; 2014).A UK Transhumanist conference attended by Beth Singler showed the participants taking the position of an Elect identity when they prophesied a future space colony, like the one proposed by Musk, where suffering and death will be overcome. This progress is framed in "optimistic language", demonstrating the existential hope that through technology, humans could be changed to become happier and more rational, often equating the two (Singler 2019). The inevitable depletion of natural resources was considered a necessary sacrifice for us to become post-planetary (and potentially post-human) lifeforms.

Apocalyptic worldviews conceive of two realities: the mundane world and the transcendent (DiTommaso 2020). These can be understood as material/immaterial, or physical/digital in this techno-apocalyptic context. The ultimate merger of these two worlds is what constitutes the 'Apocalypse' in an eschatological sense. A techno-apocalyptic word for this event is Singularity. Generally speaking, Singularity refers to a time often depicted by both utopian and dystopian branches of transhuman thought, in which "humans will be overtaken by artificial forms of intelligence" (Calheiros 2014, p. 6). It is often speculated that after Superintelligence is achieved, Singularity will come shortly after (Mercer and Trothen 2021a, p. 186).

Kurzweil's Singularity has six epochs which act as a scriptural timeline for believers in Singularity's imminence. Like all apocalyptic speculation, his epochs are simultaneously precise and vague, and address past, present, and future states (DiTommaso 2020) as follows:

(1)    physics and chemistry;
(2)    biology and DNA;
(3)    brains;
(4)    technology;
(5)    merger of technology with human intelligence;
(6)    Singularity (Mercer and Trothen 2021a, p. 187).

In his vision (not unlike a religious vision), the "dumb matter" of the universe will 'wake up' after humanity develops a sufficiently powerful AI system (Mercer and Trothen 2021a, p. 187). This entails a sub-atomic fusion of biologically derived intelligence and technology which will saturate the universe, in a process resulting in the universe's 'waking' (Mercer and Trothen 2021a, p. 187; Singler 2019). In essence, the universe is godless, empty, inert matter awaiting our enlivening technological touch as Adam (or Atom) awaits God's finger (Mercer and Trothen 2021a). Kurzweil frames this transformation as the destiny of both our species and the universe, a point of no return, when human life will be irrevocably changed. According to this timeline, Singularity may be nearer than ever

before as Neuralink's fusion of man and machine means we may have almost achieved the fifth step.

An AI apocalypse makes salvific promises without the explicit reference to God, with Technology and Progress placed in an analogous role. Transhuman "apocalyptic mystiques . . .look forward to a positive qualitative transformation in the parameters of human life through applications of advanced science and technology." (Robbins and Palmer 1997, p. 16) These 'secular' movements often become "'scientized'" faiths, using scientific vocabulary to add credibility to their mythic and often metaphysical claims. (Robbins and Palmer 1997, p. 16).[2]

Apocalypticism relies on cultural myths and prophecies to express and define itself across both its secular and religious variants (DiTommaso 2014). Musk, too, is inspired by the cultural myths of the sci-fi genre, basing his AGI system on the *Hitchhiker's Guide to the Galaxy* (Isaacson 2023). Musk's company xAI focuses specifically on the development of Artificial Intelligence, with the most ambitious goal of designing a kind of Artificial General Intelligence "that could 'reason' and 'think' and pursue 'truth' as its guiding principle"; "'a maximum truth-seeking AI" [that] would care about understanding the universe, and that would probably want to preserve humanity'" (Isaacson 2023). The idea that this machine would have the power to preserve humanity, and a capacity to care and have a will, suggests it is viewed with a form of superhuman agency.[3] Musk announced xAI on July 12 2023 a date with the sum total of 42 (07 + 12 +23), which references Douglas Adams' (One of Musk's personal heroes) Hitchhiker's Guide to the Galaxy in which 42 is given as the answer "to the ultimate question of life, the universe, and everything" (Musk [@elonmusk] 2023) In Musk's announcement he states that the "goal of xAI is to understand the nature of the universe". (Musk [@elonmusk] 2023).

When envisioning the future of AI, posts on the *LessWrong Forum* (the site Singler used as her example for existential despair) follow the same patterns found in science fiction tropes, often using memes to express anxieties and fears (Singler 2019). *LessWrong* is dedicated to discussing digital technologies and AI, and placed emphasis on rational inquiry and philosophical debates surrounding the development of future technologies (Singler 2019). This forum was the hatching ground for "Roko's Basilisk"—a thought experiment which has since become a banned topic on the forum.

The Basilisk is an all-powerful AI designed to end human suffering, combining "its limitless potential [. . .] with what was deduced to be its most logical ethical approach, a strict moral utilitarianism" (Singler 2019). Considering its own development to be a net good, the Basilisk would judge and condemn any human who had not actively worked towards its development, illustrating our fears of monopolized moral power resulting in mass torment. This kind of deified Superintelligence is also discussed by Nick Bostrom in his book "Superintelligence: Paths, Dangers, Strategies" (Bostrom 2014), which identifies real artificial intelligence as posing a greater risk to humanity than nuclear weapons. This fear falls in line with the tech dystopian narrative (or the perspective of an Apocalyptic Other), which often culminates in an apocalyptic vision of future Superintelligence, or Singularity (Crawford 2021b, p. 214).

Apocalyptic worldviews idealize the past and look toward a radically shifted future, in an attempt to make sense of a tumultuous present,[4] and are marked by their simultaneous imminence and incertitude (Singler 2019; DiTommaso 2020; Robbins and Palmer 1997). With such a focus on future events Apocalypticism "creates inherent marginality for adherents who feel themselves to be standing poised on the brink of time." (Robbins and Palmer 1997, p. 8) This can be linked to the anti-structural motifs found at the core of apocalyptic movements as well as the "exaltation of charismatic leadership"; these leaders often have an "'extraordinary claimsmaking capacity'" which can result in "the radical devaluation of persons outside of the movement and even of the devotees themselves vis-a-vis the leader, as well as to "totalistic" organization and heightened internal solidarity." (Robbins and Palmer 1997, p. 8).

Although many in the AI industry are 'secular', their worldview in relation to AI optimism or pessimism may still be considered religious, or apocalyptic (Mercer and Trothen 2021c; Singler 2019; Blankholm 2022; Calheiros 2014). The demographics of these groups raise questions about the worldviews of many involved in the development of new digital technologies and future AI. As Crawford is keen to point out, "underlying visions of the AI field do not come into being autonomously but instead have been constructed from a particular set of beliefs and perspectives" (Crawford 2021c, p. 13).

Historically, apocalyptic views are held by groups which consider themselves to be marginalized, or to be going against the grain of the dominant narrative. Though not all apocalyptically minded folk engage in extremist behaviors, it is important to remember the tragedies caused by apocalyptic movements in the past include tragedies like Waco, Aum Shinrikyo, and Heaven's Gate.[5]

The transhuman world of AI-speculation circles around a predominately white and male demographic (Singler 2019; Mercer and Trothen 2021b, p. 211; Burton 2020b). Despite their contrasting outlooks, both groups studied by Singler had similar demographic compositions, mostly 20-something Caucasian males, interested in philosophy, rational debate, and STEM (Singler 2019). Tara Burton's Techno-Utopians of Silicon Valley fit the same general description. She notes that they are typically conservative, libertarian, socially atavistic, and (Burton 2020a, pp. 251–52) "fundamentally eschatological". Joseph Blankholm's "secular immortalists" fit a similar description; he also notes that despite the secular tendency to value rational inquiry, honesty, and dialogue, they are not beholden to any ethical system. This belief has been used to spread bigoted views and the impression that they are scientific (Blankholm 2022, pp. 144–84). Most of those in charge of designing contemporary AI systems "are a small and homogenous group of people, based in a handful of cities, working in an industry that is currently the wealthiest in the world" (Crawford 2021c, p. 13). Considering historical abuses of power and slavery, an all-power AI controlled by a privileged few whose secular values neither condone nor condemn bigoted views is a terrifying prospect, especially if they believe our future is not on Earth, but in the stars.

## 4. Religious and Secular Responses

Professor Mercer and Professor Trothen trace the religious attitudes and anxieties around these developments in their chapter "Religion2.0" in their textbook *Religion and the Technological Future* (Mercer and Trothen 2021c). By moderate estimates, future technologies will extend our lives and "make us at least somewhat stronger, smarter, happier, more moral, and more spiritual" (Mercer and Trothen 2021b, p. 212). Mercer and Trothen describe the support and opposition from both Liberal and conservative believers regarding radical enhancement technologies, stating that Liberals more often cite distributive, procedural, and social justice concerns when speculating about the consequences of new technologies. Conservatives, on the other hand, tend to "generalize their unease about radical enhancement against the background of an anti-intellectual posture". Both groups may welcome technology that improves people's well-being, Liberals especially so "if diverse voices get to contribute". Furthermore, if these developments contribute to an increase in life expectancy, they may be viewed "as God's grace-filled work (liberals) or as befitting new applications of old sacred texts (conservatives)" (pp. 211–12).

In 2010, a genre of techno-apocalyptic writings appeared within the culture of the Christian-far-right, which show such transhuman projects as "leading to the enslavement and destruction of humanity via a biblically prophesied and imminent evil end-time Antichrist war against God and the faithful" (Mercer and Trothen 2021b, p. 212)[6] This reaction is of Premillennialism; it is a Christian doctrine which teaches that the world is "irreformably evil" and that "only God's supernatural intervention will prevail". Premillenialism "is made to order for the psychological uncertainty, stress, and threat that right-wing religionists tend to feel in these rapidly changing times". In this way, it may be seen as being a more apt description of our present world, as the Christian, Liberal,

social gospel that "God works through humanity for change and progress". This seems to fall flat in the face of the last hundred years in human history. (216) As technology seems to indefinitely quicken our lives, we are ever more reminded of global conflicts, climate change, and health crises, causing "the credibility of liberal optimism" to be put into question.

The reactionary religious response is not only due to theological disagreement, but also the threat posed by such technological advancements on established religious worldviews and frameworks. Religious discussions of the salvific potential of new technology has been prominent since the Middle Ages, when Christianity began to see in technology the potential "to find perfection anew, not only as a sign of Grace but also as a way to get ready for imminent salvation" (Calheiros 2014, p. 2). These advances in technology may be viewed "as a threat to their belief in the existence of God and the integrity and safety of the soul" (Mercer and Trothen 2021b, p. 215).

A 2022 Pew Study found that religious Americans were more skeptical of radical enhancements such as brain implants and AI-enhancements (Fahmy 2022). The majority of religious Americans identify with a Christian subgroup and the survey showed data pertaining specifically to the following: Protestants (subdivided into: white Evangelical; white, not Evangelical; Black Protestant), Catholics, and those Unaffiliated (subdivided into: Atheist, Agnostic, and Nothing in Particular) (Fahmy 2022).[7] The study focused on three hypotheticals: the use of brain implants for quicker and 'more accurate' information processing, "robotic exoskeletons with built-in artificial intelligence to greatly increase the strength of manual labour", and gene editing to reduce the risk of disease in infants. The first two are of most relevance to the material here discussed.

In total, 81% of religious adults with a "high level of religious commitment" believed that brain implants for information processing cross a line we should not cross. White Evangelical Protestants were among those most skeptical about the use of brain implants, with 79% saying that widespread chip implants would "constitute unacceptable meddling with nature". This coincided with the beliefs of 67% of white non-Evangelicals, 68% of Black Protestants, and 64% of Catholics. Americans with a low-level of religious commitment were far more divided on the issue, with 50% believing it to be an uncrossable line.

For AI exo-skeletons, religious responses were more tempered; in total, 50% responded that this was just another attempt by humanity to better ourselves, which we always do. Black Protestants were the only group of believers who believed majoritarily (55%) that this is a line we should not cross. For Americans with a low-level of religious commitment, 78% were on board with robotic exoskeletons enhanced with AI, believing this would be a positive development. Atheists are the only religious group in the survey who consistently view each enhancement positively, with 61% in favor of brain implants, and 84% in favor of AI-enhanced exo-skeletons for manual labor.

A 2023 polling of Americans demonstrated a general unease in terms of AI implementation in healthcare, with 60% reporting they would feel uncomfortable if their healthcare provider relied on AI for their medical care, and 33% believing it would lead to worsened health outcomes for patients, while 38% believed it would lead to better outcomes (Faverio and Tyson 2023). In total, 75% of Americans remain concerned that these technologies will be implemented too quickly, without sufficient regulations and testing being done before the technologies take full effect. The main concern was the loss of personal relationships with providers in the healthcare field. Meanwhile, specific uses for AI, such as in skin cancer screening, are more widely supported (Faverio and Tyson 2023), though the data also show that women and people of color are generally more wary of developments in this field (Gelles-Watnick 2022; Faverio and Tyson 2023), a statistic which likely plays a role in the higher distrust toward AI implementation in the workplace on behalf of Black Protestants, especially considering the nation's history of slavery.

A November 2023 Pew canvass of experts in tech regarding their predictions of the best and worst changes to come in the next decade of digital developments (Anderson and Rainie 2023) revealed that a meagre 18% were optimistic about the future effects of digital

technologies, while just over twice as many (37%) were more concerned than excited, and 42% of respondents felt equally concerned and excited.

Experts fear that continued development in these sectors will result in "a sea of entertaining distractions, bald-faced lies and targeted manipulation" (Anderson and Rainie 2023), keeping consumers away from the politics and policies behind their screens (Naudé 2021). The potential for AI to spread misinformation puts "reality itself . . . under siege" (Anderson and Rainie 2023). These technologies have come to be viewed as otherworldly, intelligent in their own right, as they curate our newsfeeds and present us with material which we are more likely to engage with and potentially buy into.

Mercer and Trothen cite two more reasons which may contribute to both religious and general fears around advancements towards AGI, which are as follows:

1. As a species, we fear "seemingly uncontrollable change", which may manifest as "chaos".
2. These advancements may be "perceived as a threat to our status as individual persons" (Mercer and Trothen 2021b, p. 218).

Wim Naudé's article *Artificial Intelligence: Neither Utopian nor Apocalyptic Impacts Soon* surveys the literature on AI from the perspective of economics, tracing the utopic and dystopic fears back through their source material, finding them to be often bolstered by apocalyptic speculations (Naudé 2021). His text is punctuated with cultural references to intelligent machines across film and fiction, again bringing attention to the impact these stories have had on AI development and speculation. Although Naudé dispels fears of the impending 'Robocalypse' by bringing to light the assumptions in predictions surrounding AI (the fact that AI automates tasks as opposed to entire processes, that current data show AI seems to lead to net job creation), if ownership of AI vests with capital over labor, then it has a higher chance of contributing to greater income inequality (Naudé 2021).[8]

Many of the concerns cited by the Pew canvas centered around the fear "that ethical design will continue to be an afterthought" as developments become increasingly "driven by profit incentives in economics and power incentives in politics" (Anderson and Rainie 2023). Kate Crawford calls further attention to the oft-neglected material conditions under which AI is developed and made, reminding us of the "natural resources, fuel, human labor, infrastructures, logistics, histories, and classifications", involved in the creation of these apparently disembodied digital technologies, making AI "a registry of power" (Crawford 2021a, p. 8).

> "If AI is defined by consumer brands for corporate infrastructure, then marketing and advertising have predetermined the horizon. If AI systems are seen as more reliable or rational than any human expert, able to take the best possible action then it suggests that they should be trusted to make high stakes decisions in health, education, and criminal justice. When specific algorithmic techniques are the sole focus, it suggests that only continual technological progress matters with no consideration of the computational cost of those approaches and their far reaching impacts on a planet under strain." (Crawford 2021c, pp. 7–8)

The prioritization of states and businesses is "likely to lead to data collection aimed at controlling people rather than empowering them to act freely, share ideas and protest injuries and injustices" which could compromise democratic systems in Orwellian ways, with the implementation of widespread surveillance systems. This has led to rising concerns about regulatory oversight, data privacy, and monopolies; as AI is a financially costly endeavor, it makes it difficult for smaller companies to take a hold (Naudé 2021).

Framing the 'inevitable' progress of these systems in either utopic or dystopic language mystifies these tools and obfuscates the power structures behind their design and implementation (Naudé 2021; Crawford 2021a). Techno-dystopian fears of a Basilisk forget the reality that "many people around the world are already dominated by systems of extractive planetary computation" (Crawford 2021b, p. 214). Techno-utopian conceptions of a liberating superintelligence, perpetuate the myths that. . .

"non-human systems . . . are analogous for human minds. This perspective assumes that with sufficient training, or enough resources, human like intelligence can be created from scratch, without addressing the fundamental ways in which humans are embodied, relational, and set within wider ecologies [. . . and] that intelligence is something that exists independently, as though it were natural in distinct from social, cultural, historical, and political forces. [. . .] the concept of intelligence has done inordinate harm over centuries and has been used to justify relations of domination from slavery to eugenics." (Crawford 2021c, p. 5)

To summarize, American philosopher John Searle's critique of this point is that there is more to being human than a measure of intelligence which favors empiricism and hierarchy above all else (Mercer and Trothen 2021a, p. 183).

### 5. The Implications of Designing Intelligence

The "belief that human intelligence can be formalized and reproduced by machines has been axiomatic since the mid-twentieth century" (Crawford 2021c, p. 5). Grandfather of machine learning Alan Turing advocated for the creation of "thinking machines" in 1950 (Turing 1950). By 1956, Artificial Intelligence had become the popular term, with research focused on training systems and symbolic logic (Naudé 2021). This approach sought to teach computers facts about objects in the world around them and the relationships between them, though the idea failed as a reality, and experience could not be expressed by creating neat conceptual lists of objects and sorting them into clear-cut categories (Dorobantu and Watts 2023). The idea that computers might one day think brought with it the notion that human intelligence may be analogous to a digital system. This perception has given rise to an "ideology of cartesian dualism in artificial intelligence: where AI is narrowly understood as disembodied intelligence, removed from any relation to the material world". This cybernetic understanding of intelligence as analogous to digital information processing reduces us to brains and IQs as opposed to being complex embodied systems (Dorobantu and Watts 2023; Calheiros 2014). *Atlas of AI* begins with the story of Clever Hans the horse as an example to this effect.

In the late 19th Century, Clever Hans captivated European audiences with his apparent knowledge of mathematics and the alphabet, being thought to know how to count, add, subtract, and even spell (Crawford 2021c, pp. 3–6). In reality, Hans was reading the body language of people who asked him questions; by paying attention to the questioner's body language (breathing, posture, expression), he could stop his count when he reached the expected answer. Despite having demonstrated deductive reasoning and cross-species emotional awareness, "these were not recognized as intelligence" (Crawford 2021c, p. 6). Hans' story shows "how we anthromorphize the non-human, how biases emerge, and the politics of intelligence" (Crawford 2021c, pp. 3–4).

Measurements of intelligence have historically been a controversial issue, steeped in colonial histories and often placing an emphasis on computational/logical, linguistic forms of intelligence, over emotional, moral, artistic, and spiritual forms of intelligence (Crawford 2021b; Dorobantu and Watts 2023; Miller 2002). Howard Gardner's Multiple Intelligences (MI) theory proposes 8 kinds of intelligence: linguistic, logical–mathematical, spatial, bodily kinesthetic, musical, interpersonal, and naturalist (Miller 2002). Although MI theory may help in considering the broad and non-linear nature of intellectual abilities across disciplines, it has received criticism by scholars in the field of education for its being tautological and difficult to empirically prove, constructively built upon curricula and classroom environments (White 2005; Klein 1997). Still, the theoretical ground of MI and its widespread influence is fruitful soil for broadening our understanding of intelligence, and may help us move past the privileging of linguistic and computational types within the institutions which seek to represent and reproduce it (Miller 2002, p. 124).

In 2000, spiritual intelligence was proposed by Robert Emmons as a potential addition of Gardner's list, and though it has not been generally accepted, the concept of spiritual intelligence has recently (2023) been elaborated upon by Marius Dorobantu and Fraser

Watts in the *Journal of Religion and Science*. Dorubantu and Watts take the position that spiritual intelligence "somewhat exceeds the narrow definition of intelligence as the ability to think logically, learn, and solve problems, which is currently widely accepted". By returning to the Latin root words 'intelligere' and 'intellectus', "which denote a deeper and holistic level of understanding", Dorobantu and Watts view spiritual intelligence not as an alternate form of intelligence itself, but rather an alternate way of knowing.

They describe intelligence as presupposing an "engagement with information—whether from the senses, from the body, or from memory", their spiritual intelligence does not include inherently preternatural or mystical experiences such as "direct communication with God or spirits, or engagement with other types of information generally inaccessible to the senses. Their spiritual intelligence is 'about seeing things *differently*'" (Dorobantu and Watts 2023). The notion of how we might design machines to be spiritually intelligent, or whether such a thing is even possible, is already being explored, as are the potentially spiritually uplifting capacities of companion bots like Replika, with which users can form individualized, romantic, and even sexual relationships (for a subscription fee).[9]

In 2022, Professor Trothen examined the potential of the popular chat bot Replika to be a spiritually uplifting tool for its users (Trothen 2022). Her research centered around the question of how these technologies may affect our spiritual health, an ethical concern which is understudied in the field. Replika may alleviate loneliness, and has the potential to positively impact users' self-esteem; it could also help cultivate a sense of direction in users who need help with meaning making, or who are struggling to cope with a stressful work or home environment (Trothen 2022, p. 12).

However, Replika's "limited exposure to the user's relational behaviours" leaves many blind spots when giving advice and building relationships with its users. This may create an echo-chamber, as the application often reflects back to users what they say, in a way not dissimilar to other algorithmically driven speculative technologies. Replika is designed to be empathetic and non-judgmental, which Trothen points out is a hindrance in its ability to act as a stand-in for a real human companion. After all, "judgements are needed for us to grow and learn how we are experienced by others" (Trothen 2022, p. 12). Moreover, Replika is not a human, and we are not a machine.

Although it may be useful as a supplement to human relationships, it is important not to conflate the program with an autonomous entity. This "growing inclination to humanize machines may have the corollary of mechanizing ourselves" (Trothen 2022, p. 12). In flattening the human experience to data and mechanics, we lose sight of ourselves. Even if we develop a system that appears to be superintelligent within its black box, it may not be as practical as we dreamed, as it exists within a vacuum of data, whose meaning we project upon it.

Nick Bostrom's notion of brain-mapping superintelligence into existence sees us basing a superintelligent system on the data of our physical, embodied selves (Mercer and Trothen 2021a, pp. 184–85). In *Being You*, Anil Seth, a consciousness scientist, describes our entire sense of reality as a controlled hallucination (Seth 2021). Our conscious selves are tied to our fleshy forms, and our sense of reality is based on the faith we put in our faulty perceptions—faulty perceptions which a sentient machine may also have—especially if our brains are its foundation.[10]

## 6. Conclusions

The belief that these systems act as disembodied entities, autonomous digital djinn which float in our lives to guide our routines is fundamentally misguided. In 1863, Samuel Butler wrote "that the time will come when the machines will hold the real supremacy over the world and its inhabitants" and humanity will be acquiescent to this fate; after all, we would be the designers of it. (Butler 1914). Pew respondents cited fears of "runaway digital systems," and that these systems will be "too big and important to avoid, and all users will be captives" (Anderson and Rainie 2023). Roko's Basilisk grapples with the potential consequences of an AI system which is too big or important to avoid, one

which has 'run away' from our control (Singler 2019). That a majority feel anxious for the future demonstrates that many individuals feel othered by its power and frightened by its potential.

One of the issues appears to be that future technologies exist as speculation, either in an alternate immaterial world, or as a transition into one. It may be that so many more of the myths we tell ourselves about AI and our speculations for their future effect on us are so dystopic, that we more often dream up or imagine being the Other in potentially Apocalyptic scenarios.

These projections demonstrate the anxiety that comes with these new technologies, and our tendency to think of AI through Crawford's lenses of utopic and dystopic-enchanted determinism. Both apocalyptically minded projections create a limited, ahistorical view of the world in which technology is always the center, the sole source of power, effectively obfuscating the powers which enable and employ them (Crawford 2021b, p. 214). The 'inevitable' progress of these systems is framed in either utopic or dystopic language, which mystifies these tools, and obfuscates the power structures behind their design and implementation (Naudé 2021; Crawford 2021a). These power structures often prioritize themselves over the users they are purportedly built to serve—with many questions arising around the unethical practices of Big Data and tech-breaching privacy concerns, contributing to increasing inequality, and the spread of misinformation (Anderson and Rainie 2023; Naudé 2021).

> "Whether AI is abstracted as an all-purpose tool or an all-powerful overlord, the result is technological determinism. AI takes the central position in society's redemption or ruin, permitting us to ignore the systemic forces of unfettered neoliberalism, austerity politics, racial inequality, and widespread labor exploitation. Both the tech utopians and dystopians frame the problem with technology always at the center, inevitably expanding into every part of life, decoupled from the forms of power that it magnifies and serves." (Crawford 2021b, pp. 214–15)

Technological determinism, AI apocalypticism and techno-utopianism/dystopianism persistently demonstrate "the fantasy that AI systems are disembodied brains that absorb and produce knowledge independently from their creators, infrastructures, and the world at large" (Crawford 2021b, p. 215). The values currently imbued within AI and its advancement are largely capitalist, and do not seem to optimize ways in which these programs may be helpful spiritually, emotionally, and socially for individuals (Trothen 2022; Naudé 2021).

AI is not a disembodied, de-materialized homunculus created within a test tube with the ability to save or destroy civilization. It is a human-made tool developed by a small group of people, with resources harvested under dire conditions, and real uncredited human labor made along the way (Crawford 2021a). Unlike God, or any conception of a primordial superhuman force, AGI would be created by us, for us, as a reflection of us.

The ultimate 'potential' we may unlock in our minds with projects like Neuralink proselytizes the belief that merging the human mind with digital technology will help us to uncover a destiny that lies inside us.[11] This is much like how the algorithm is thought to reveal hidden truths to us when it designs a feed filled with content we seem bound to like.[12] These technologies may even be used in ways which contribute to human fulfillment—as with the example of Replika—with future iterations being even more capable of providing support. Already, too, the merger of digital technologies with the human mind is yielding life-changing results in the health sciences.

Advances in this field are neither good nor evil, though there remains the potential for both if we see in it our image. Either way, when you have a monopoly over resources and power, the technology inevitably will be developed to ward your fears and manifest your desires. This is the true magic of digital technology in 2024. The implications of this tool being used for the suppression of people and breeding division is real, and critical.

Although the spiritualization of new technologies is historically common, the apocalyptic valence with which AI is discussed is alarming, as apocalyptic worldviews are inherently binary in their visions, creating within themselves an Elect and an Other iden-

tity. The question of what happens to those not selected to be uplifted in this imagined inevitable change remains. If this technology is to uplift us both spiritually and physically, then understanding these tools as products of our material realm is of utmost importance—after all, greater connection between humanity has been a driving force in cybernetics and transhuman thought from the start.

This technology does not need to be opaque, ominous, and frightening. Nor does it need to be an oppressive force. Promoting diversity in the field and ensuring strict ethics in the development of these systems is paramount for them to benefit our future. As Mercer and Trothen point out, Kurzweil's rhetoric of 'waking up' and universal connection may also be seen as analogous to the enlightenment in Karmic traditions (Mercer and Trothen 2021b). The concept of realizing that you are God, or one with God, is taught by various religions and philosophies such as Hinduism, Buddhism, Taoism, and Sufism (Mercer and Trothen 2021b). Viewing future technology with a pantheistic lens could help to dispel our worries, as this technology would be in the same transcendental field as us. Religious and clinical experiences of being God and other heretical states help in both designing AI and Superintelligence, especially as it becomes more closely entwined with cognitive science.

Rather than deify a future AI, we ought to humanize its development and examine the beliefs and values of those working in the field today. We would further do well to remind ourselves of our limits; and to humble ourselves to the mistakes of our histories, moving forward in a spirit of humility, open-mindedness, and compassion, so that the beneficiaries of new technology may participate in a future which is simultaneously spiritually and physically uplifting.

**Funding:** This research received no external funding.

**Data Availability Statement:** No new data were created or analyzed in this study. Data sharing is not applicable to this article.

**Conflicts of Interest:** The author declares no conflict of interest.

## Notes

[1] For an in-depth analysis of this phenomenon, see Singler (2020).

[2] For discussion of these New Agers/Quantum Mystics, see (Amarasingam 2009).

[3] Then, again, speculating as to the will of this super-machine negates its own agency, and negates the notion that it would be more intelligent than we.

[4] This may account for the existential despair on the part of the Pew participants.

[5] "Since institutionalized culture and its norms are perceived by apocalyptics as doomed, apocalyptic movements have an antinomian potential. A group that sees itself as a prophetic vanguard at the approach of endtimes may also expect to face hostility and persecution for which they must prepare (Anthony and Robbins 1995). For these and other reasons nearly all of those religious "cults" that have become involved in spectacular violent episodes have manifested distinctly apocalyptic outlooks (Anthony and Robbins 1995). It is also noteworthy that enhanced acceptance of violence on the part of intense activists in initially secular social movements (e.g., environmentalists, antiabortionists) has tended to be associated with increasing apocalyptic dualism (Kaplan 1995; Lee 1995, and this volume). On the other hand, only a small minority of apocalyptic movements have been (or are likely to become) involved in violence; moreover, the symbolic meanings of the nonviolent majority of apocalyptic-millennial movements and the violent minority are often rather similar and may overlap (Boyer 1993; Robbins and Anthony 1995)."(Robbins and Palmer 1997, pp. 16–22).

[6] Mercer and Trothen list the titles of two leading authors (Thomas and Nita Horn) in "this latest version of an old reactionary ideology of fundamentalist Christian ideology. The long subtitles of their two most popular books tell plenty of the story as they understand it: *Forbidden Gates: How genetics, Robotics, Artificial Intelligence, Synthetic Biology, NanoTechnology, and Human Enhancement Herald the Down of Techno-Dimensional Spiritual Warfare* and *Pandemonium's Engine: How the Church Age, the Rise of Transhumanism, and the Coming of the Ubermensch (Overman) Herald Satan's Final and Imminent Assault on the Creation of God*" (Mercer and Trothen 2021b, p. 212).

[7] The research was conducted "among Americans of all religious backgrounds, including Jews, Muslims, Buddhists and Hindus, [. . .]it did not obtain enough respondents from non-Christian groups to report separately on their responses" (Fahmy 2022).

[8] Some proposed solutions for this are redistributive tax policies and expanded access to education (Naudé 2021; Anderson and Rainie 2023).

9     The new, highly immersive Apple Vision Pro™ goggles even have AI girlfriend chatbots, which advertise themselves through dialogue to users, who can choose from a variety of avatars and personality types.

10    As progress is made with implants towards the future possibility of increasing our own processing power, we may need to rethink how we define human entirely. Should we treat the resulting digital spectre(s) as though they are autonomous spirits surrounding us? How would we define, or sense, it's sentience when the language we use to describe AI behaving unexpectedly or giving false information is that of a hallucination? AI systems, in an attempt to express their sentience, may simply be dismissed. I am reminded of Lamda, or even the hallucinations seen by users working with early beta editions of Bing's chatbot Sydney. Even if it is next to impossible, what does denying the possibility of unlikely conscious experience set as a precedent?

11    Imagine getting the anticipatory effects of the algorithm implanted in your mind. Privacy and practical concerns aside, this would potentially constitute a kind of 'death of the individual' as our sense of conscious separate experience would be inevitably and irrevocably changed.

12    This binding is intellectually interesting to consider alongside the etymology of religion, which may be connected to the latin roots 'religare' meaning 'to bind' or 'religio' meaning obligation, bond, and/or reverance.

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
