# Peer review of "There’s a Basilisk in the Bathwater: AI and the Apocalyptic Imagination"

_religions, doi:10.3390/rel15050560_

Round 1

Reviewer 1 Report

Comments and Suggestions for Authors

The central question of this essay is whether there are fruitful parallels between the history of apocalyptic literatures and discussion surrounding the advent of AI. Clearly there are: many experts in computing seem to regard the coming of AI along the lines of the coming of a god. The discussion is somewhat superficial, as it consists mainly in pointing out parallels and echoes in the discourse, but the observation is definitely worth making, and the paper presents an opportunity for religious scholars to find their relevance in one of the hottest developments in our society today. The literature that is cited is relevant, and I noticed no further sources that are necessary to consider. The discussion could be improved by perhaps focusing on only a few discussions of AI (e.g., Kurzweil's The Singularity is Near) and noting explicit parallels between that discussion and specific eschatological tropes or themes. In short: less would be more. But as it is, the essay is still worth placing in a public space so that other scholars might be able to press the ideas further into their own realms of specialization.

Author Response

To Reviewer 1 of “There’s a Basilisk in the Bathwater: AI and the Apocalyptic Imagination” the special issue of Religions “Theology and Science: Loving Science, Discovering the Divine”,

April 22, 2024

Thank you for reviewing my essay “There’s a Basilisk in the Bathwater: AI and the Apocalyptic Imagination”, which I have now submitted for your second (and hopefully final), review.

Following your suggestions, I have tried to narrow my focus and more clearly note the explicit parallels between topics. “Less is more” is usually something a struggle with, and is in part why I have decided to include footnotes this time around. Your suggestions were especially useful in my revisions of sections 2 and 3 (which now discuss the parallels between transhuman thought and algorithmic enchantment, and the transhuman worldviews to apocalypticism respectively).  Here is an outline of the changes made in this manuscript after careful consideration of the comments and suggestions:

  1. Revised Introduction, mostly slight rephrasing to lower the word count, as well as some reframing which seeks to streamline and narrow the scope of the essay. I have tried to more clearly connect the religious attitudes towards the development of superintelligence. I have also reordered the information in a way which hopefully helps the connection of these seemingly disparate ideas is theoretical/conceptual discussion. 
  2. Section 2 now connects contemporary attitudes towards algorithmic ways of knowing to transhuman thought. I have added more references to the way in which algorithmic technologies are spiritualized in the first paragraph, as well as specified more precisely their link to conspiratorial thought. This section also now explicitly connects Calheiros’ discussion of Global consciousness to the wider topic of superintelligence/the singularity. The other revisions in this section are mostly the shortening/ combining of phrases.
  3. Section 3 now more directly discusses the similarities between Apocalypticism and some Transhuman projects, as well as the technological advances which the future may hold.
  4. Section 4 is mostly new material, inspired by the helpful comments of reviewer 2, and discusses the religious (mostly Christian) and secular responses to the emergence of new technologies.
  5. Section 5. elaborates on the limitations of developing AI further by problematizing the datafication of ‘intelligence’ and the belief that the human mind may operate like a digital system. Two paragraphs on Intelligence theory and Spiritual Intelligence have been added.
  6. As a result of the restructuring, the conclusion was reworked and condensed, with some material being removed.
  7. I have also added footnotes.

Your comments have been invaluable and have helped to improve the quality of this research, and will contribute to the betterment of my scholarship overall.

With much gratitude,  

The author.

Reviewer 2 Report

Comments and Suggestions for Authors

The reviewed work discusses the issue of threats related to the uncontrolled development of artificial intelligence, deeming the advent of General Artificial Intelligence (AGI) as a pivotal moment in this development. AGI is a hypothetical form of artificial intelligence that has the capability to learn, reason, understand, and act in a manner fully comparable to that of the human mind. In other words, AGI could perform any intellectual task a human can. This distinguishes AGI from narrow artificial intelligence, which is designed and trained for specific tasks (e.g., image recognition, language translation, chess playing). AGI poses a significant theoretical and technical challenge, as it requires not only advanced machine learning and data processing capabilities but also an understanding and modelling of the highly complex human thought processes, including emotions, intuition, and morality. To date, AGI remains within the realm of theory and research, with the majority of currently available AI systems being specialized, capable of performing specific tasks within certain domains. The literature, however, often speaks of an even more complex form of artificial intelligence known as superintelligence. Superintelligence refers to a hypothetical form of artificial intelligence that surpasses all human minds in cognitive abilities, including both the speed of information processing and the depth of understanding. In the context of artificial intelligence, superintelligence would mean a system or machine whose intellectual level exceeds the highest possible human achievements.

Various concepts and theories exist on how superintelligence could be achieved. One hypothesis is the creation of AGI capable of self-improvement, i.e., the ability to teach itself and improve its own algorithms. Such AGI could rapidly surpass human intellectual capabilities and become superintelligent.

Superintelligence is a topic widely discussed in philosophy, ethics, computer science, and other fields. There are both enthusiasts who see in it the potential to solve humanity's greatest problems and concerns regarding control over such a powerful form of artificial intelligence and the consequences that could result from excessive reliance on it. The author of the reviewed work seems to advocate for the latter option, as indicated by the title, which alludes to apocalyptic imagination. This does not directly refer to biblical theology but uses the word "apocalyptic" to denote a situation associated with catastrophic events, or even the end of the known world. The term "apocalyptic" is typically used to describe something very negative, destructive. For instance, one can speak of an "apocalyptic scenario" to describe a very pessimistic vision of the future or a possible course of events leading to disaster. Thus, the work is not a study in theology or the philosophy of religion, but a general essay introducing topics related to the development of this brilliant invention, artificial intelligence. By calling AI "brilliant," I reveal my own view on AI, slightly different from the thought pursued in the reviewed work. Namely, I consider AI a very precious gift to humanity. I am aware that it is simultaneously a task for humanity, as the development of artificial intelligence opens the way to many abuses. This problem has not gone unnoticed by the creators of AI-based tools. As AI develops, actions are taken to at least partially prevent abuses. Hence, my view on this matter is less "apocalyptic" than that of the article's author. However, I firmly agree with the statement that the development of AI should be accompanied by the advancement of humanities in this field, and I believe the reviewed work constitutes a valuable contribution to this development.

However, the theory according to which artificial intelligence is elevated to a spiritual level does not convince me. The classic Searle's room experiment shows that AI will never be what we call intelligence in the philosophical sense (primarily because AI will never acquire self-awareness). Even if it passes the Turing test with the highest grade, it will always be only a simulation (though a very perfect one) of human intelligence. Any "spiritualization" of artificial intelligence reminds me of the Marxist law of the transformation of quantity into quality, proclaiming that when matter reaches a certain degree of complexity, spirit emerges from this matter, i.e., a qualitative change occurs. However, even the most complex AI-based computer system will never dematerialize. For this reason, superintelligence will never be equated with an absolute being, i.e., God (unless only in an analogous sense). It seems unlikely that anyone would ever pray to superintelligence as they do to God. I understand, however, the author's intentions and am aware of the rhetorical figure he employs. If I may suggest, the author could consider introducing a religious theme, i.e., he could try to contemplate whether the potential emergence of superintelligence would lead to a revision of the classical concept of God.

Author Response

Thank you. 
